# Educational Intervention to Decrease Justification of Adolescent Dating Violence: A Comparative Quasi-Experimental Study

**DOI:** 10.3390/healthcare11081156

**Published:** 2023-04-18

**Authors:** Jesús Alberto Galdo-Castiñeiras, Juan José Hernández-Morante, Isabel Morales-Moreno, Paloma Echevarría-Pérez

**Affiliations:** 1Health Sciences PhD Program, Universidad Católica de Murcia, 30107 Murcia, Spain; jagaldo@ucam.edu; 2Faculty of Nursing, Universidad Católica de Murcia, 30107 Murcia, Spain; imorales@ucam.edu (I.M.-M.); pechevarria@ucam.edu (P.E.-P.)

**Keywords:** intimate partner violence, violence justification, health education, adolescent, psychosocial intervention, nurses, public health

## Abstract

Adolescent dating violence has become a public health problem because of the associated high morbidity and mortality rates. Despite social awareness about dating violence, the high justification of violence among adolescents is one of the main risk factors for both perpetration and victimisation. Therefore, the objective of the present work was to evaluate the effectiveness of an educational intervention in reducing the justification of violence in adolescent dating. A quasi-experimental, longitudinal, prospective study with a control group was conducted. The study was carried out in six different schools in the Region of Murcia (Spain), and the participants were 854 students aged 14–18 years. The educational intervention was focused on reducing the justification of adolescent dating violence and consisted of 9 weekly 1 h group sessions. The Justification of Verbal/Coercive Tactics Scale (JVCT) and the Attitudes About Aggression in Dating Situations (AADS) survey were administered at baseline and at the end of the intervention in order to determine the justification of psychological and physical violence, respectively. At baseline, the justification of physical violence was at a medium-to-high level in boys (76.8%) and girls (56.7%), whereas psychological violence was much less justified. Concretely, female psychological violence was justified by 19.5% boys and 16.7% girls, while male violence was justified by 19.0% boys and 17.8% girls. After the educational intervention, a significant decrease in physical violence justification, especially in the AADS dimension of female aggression, was observed. The effect of the intervention was especially evident in psychological violence justification: a statistically significant difference was observed in the JVCT scores of boys (−6.4 and 1.3 points in the intervention and control groups, respectively; *p* = 0.031), but not of girls (*p* = 0.594). In conclusion, the educational intervention was adequate to reduce the justification of dating violence among the participants. It may provide adolescents with the skills and resources necessary to confront and resolve conflicts in relationships in a non-violent manner.

## 1. Introduction

In 1996, the World Health Organization declared that violence in intimate partner relationships is no longer a hidden problem but a real health concern. The main reason for this is the high morbidity and mortality rates associated with this kind of violence, with incidence and prevalence rates exceeding those of cervical or breast cancer [1,2]. Despite this, we cannot ignore the fact that there is still a certain invisibility factor [3,4] to the problem, which implies that the true incidence rates may be even higher.

According to the most recent Spanish report on violence against women [5], 38% of girls aged between 16 and 24 years have suffered physical, psychological and/or sexual dating violence. Similar data are found in European and other countries [6,7]. Among those who experienced dating violence, 26% of girls and 15% of boys first experienced partner violence before the age of 18 years [8]. Early onset of dating violence predicts an increase in the severity and chronicity of partner violence [9]. Shorey et al. [10] found that the greatest risk of onset of physical dating violence was at or before ages 15 to 16 for females and at or before the age of 18 for males. Fernández–González et al. [11] noted a peak in physical dating violence between the ages of 16 and 17 years. Overall, these data highlight the magnitude of the problem in adolescent dating. In a systematic review, Rubio–Garay et al. [9] emphasise the seriousness of this issue by reporting the figures of violence committed against a partner: 40.3% of boys and 41.9% of girls had committed physical violence; 95.3% of boys and 97% of girls had committed psychological violence; and 58.8% of boys and 40.1% of girls had committed sexual violence. Zhang et al. [12] reported that in recent years, an increase in aggressiveness was noted among young people, favouring the use of physical and especially psychological violence.

Paradoxically, although educational programmes on equality and the prevention of dating violence have increased [13,14], the incidence of dating violence is also increasing [6]. This, among other causes, is due to social norms that condone or justify violent attitudes in adolescents and which drive them to consider violence a good and effective method of partner conflict resolution [15,16], and this, in turn, encourages and increases the bidirectionality of dating violence in both sexes [17]. Several authors report that attitudes towards violence can be a predictor of the future development of violence [15,18,19]; thus, the greater the level to which a person justifies the use of violence in a partnership, the greater the risk of not only committing it in their relationship but also suffering it [20,21,22]. However, previous interventions carried out to reduce violence, like those previously developed by McFarlane et al. [23] and Petersen et al. [24] have been focused on general aspects of dating violence and not on more specific aspects such as the justification of violence, which could partially explain the low effectiveness of these previous interventions.

For all the above reasons and given the low effectiveness of educational interventions aimed at the direct eradication of violence, the objective of the present study was to conduct a nursing intervention to prevent the justification of dating violence in educational contexts, as part of the development of the functions of the school nurse. The intervention focused on reducing the justification of dating violence in young people, which in turn would reduce the risk of violent attitudes that have serious consequences for life in general and for health in particular.

## 2. Materials and Methods

### 2.1. Design

A quasi-experimental, randomised, longitudinal, prospective study was conducted. Eight schools were initially contacted; two declined to participate, claiming that the intervention consisted of too many sessions. Finally, the study was conducted in six schools in the Region of Murcia (Spain) between January 2018 and March 2020.

### 2.2. Participants

The sample size required for the study was calculated taking into account the fact that the adolescent population in Murcia, aged 14–18 years, was 80,108 in 2018. A sample size of 383 students was adequate to obtain a 95% confidence interval. Nonetheless, for this study, 938 secondary school students were selected. Permission to participate in the study was obtained from the management teams of the six schools as well as from parents or legal representatives. The exclusion criteria were refusal of consent (n = 3) and not understanding Spanish. Participants in the control group were offered the intervention at the end of the study. Participants who did not complete the educational intervention (n = 24) or did not complete the tests (n = 57) were excluded from the statistical analysis. Finally, 854 students completed the study, and their data were considered in the final dataset. This student sample was randomised into two groups (Figure 1): the intervention group (410 students) and the control group (444 students).

Participation was anonymous and voluntary and required the express provision of written informed consent. Participants could withdraw their consent at any time. Data were collected exclusively on the age and sex of the participants. Approval was obtained from the Ethics Committee of the Catholic University of Murcia (CE051711).

### 2.3. Educational Intervention

The study intervention was designed for this study as a part of the European project Vida Youth (https://vidayouth.wordpress.com/2016/11/06/the-project/, accessed on 12 January 2023) [25]. The general objective of the project was to prevent violence among young people due to the lack of awareness of the nature of romantic relationships. In particular, VIDA YOUTH pursued the following specific objectives: to exchange best practices in the field of the prevention of youth violence between partners; to raise awareness on the issue of violence among young people in contact with youth organizations, teachers, educators, students and young people in general; and to strengthen the non-formal education courses of schools [25].

A face-to-face educational intervention of three months was delivered in six educational establishments for the prevention of teen dating violence. It focused on reducing the levels of justification of violence and consisted of 9 weekly group sessions of 1 h each. The sessions were held during high-school time, in groups of 25–30 students per session. The academic mentor responsible for every group was present at all sessions. At the beginning of each session the main concepts were recalled.

The teaching program was made up of sessions that addressed the topics: awareness, focused on raising awareness of the problem of violence in adolescent dating; causes and risk factors of violence in young couples, where several risk factors were discussed, both for victimization and for the perpetration of violence; consequences of violence, with sessions focused on the physical, emotional and psychological consequences both to the victim and to the aggressor; tolerance and values, focused on the respect of individual traits; equal treatment of boys and girls, with sessions based on an equal dignity for all people; communication problems, focused on problem-solving skills from a non-violent perspective; communication skills, focused on improving communication skills in the couple; prevention of aggression, focused on learning the ability to establish boundaries in the couple; and the influence of new technologies, with sessions based on the prevention of online violence.

In order to aid the transmission and assimilation of the material, in addition to the talks, group activities were carried out, and audio-visual aids were used in the form of videos, presentations and other media. In all of the sessions, participatory methods were used with the students, allowing them to interact, raise doubts and narrate their own experiences, which favoured the acceptance of the programme and its objectives, as previously described by Emerson et al. [26].

### 2.4. Primary Outcome

In order to assess the effectiveness of the educational intervention, changes in participants’ scores on the Attitudes About Aggression in Dating Situations (AADS) questionnaire (for justification of physical violence) and on the Justification of Verbal/Coercive Tactics Scale (JVCT) questionnaire (for justification of psychological violence) were considered as the main outcomes. As secondary outcomes, several sociodemographic data were also evaluated.

### 2.5. Data Collection Instruments

Before the start of the educational intervention, the sociodemographic data and those referring to the participants’ partner relationships were obtained through a questionnaire specifically developed for the present study. In addition, two tests (AADS and JVCT) were administered in order to determine the level to which participants justified the use of dating violence as well as to detect their lack of knowledge and distorted attitudes about dating. At the end of the intervention, the tests were administered again to measure changes in the levels of justification of violence and assess the effectiveness of the intervention.

Attitudes About Aggression in Dating Situations (AADS): The AADS scale (developed by Slep et al. [27]) assesses the level of justification of physical violence in dating situations, contextualising, over 10 items, different situations in which physical aggression materialises in a bidirectional way: aggression committed by a boy against a girl (male aggression, 4 items), aggression committed by a girl against a boy (female aggression, 4 items) and aggression between peers (peer aggression, 2 items). Participants use a six-point Likert-type scale, from 1 (total agreement) to 6 (total disagreement), to express their level of agreement or disagreement with physical aggression committed by a boy or a girl, either as a response to aggression received previously (physical or psychological) or as a reaction in a context of jealousy. The higher the score, the lower the level of justification of physical aggression by the participants. The AADS was validated in Spain in 2011 by Muñoz–Rivas et al. [28] (α = 0.83).

The Justification of Verbal/Coercive Tactics Scale (JVCT): The JVCT [27] measures the level to which participants justify psychological violence committed by boys and girls, based on three dimensions: male or female verbal aggression (4 items), male or female dominance tactics (4 items) and male or female jealous tactics (4 items). The scale consists of 12 items, which each respondent must evaluate twice: first to indicate their level of justification when the aggression is committed by a girl, and then, when it is committed by a boy. Participants use a five-point Likert-type scale (from 1 = never justified, 5 = justified on many occasions), and the lower the score obtained, the lower the level of justification of psychological aggression shown by the participants. The JVCT was validated in Spain in 2011 by Muñoz–Rivas et al. [28] (α = 0.90).

### 2.6. Data Analysis

An initial descriptive analysis was carried out before the intervention in order to examine the level to which participants justified physical and psychological dating violence. Subsequently, an inferential analysis was carried out to verify whether there was an association between the different variables in our study (age, sex and justification). Finally, we analysed the changes in the variables at the beginning and the end of the intervention to ascertain the level of effectiveness of the educational intervention. Taking into account the ordinal nature of the main variables, the non-parametric Mann–Whitney *U*-test was used to determine the baseline differences in the level of justification. The differences between the pre-test and post-test scores were measured with the Wilcoxon test for related samples. To compare intervention effectiveness, mean changes (estimated as post-intervention minus baseline value) between intervention and control groups were also analysed by the *U*-test. Effect sizes were expressed as r-value, according to the American Statistical Association statement on *p*-values [29]. Finally, to better interpret this effect size, the difference-in-differences equation according to the following multivariate linear specification, was estimated:*Yt* = *β*0 + *β*1*t* + *β*2*re* + *β*3*re*
*t* + *ϵt*
(1)
where *Yt* refers to the scores of the subjects of the variable of interest (AADS and JVCT dimensions) at baseline and post-intervention; *ϵt* is the error term and is usually assumed to be correlated within the same subject. The coefficient *β*1 is the slope of the line for the control group. The coefficient *β*2 is the initial difference between the two groups (effect of belonging to one treatment group or another). The coefficient of interaction *β*3 is the measure of the effect size of the difference in differences. The data were analysed with IBM SPSS (V26) and R (version 4.2.0, package: foreign), and the level of statistical significance was set at *p* < 0.050.

## 3. Results

### 3.1. Characteristics of the Participants

Fifty-two percent (n = 450) of the participants were boys. The mean age of the participants was 15 years, and it was similar for boys and girls (*p* = 0.940). In the intervention group, 44.7% (n = 183) of the participants were girls; in the control group, 49.8% (n = 221) were girls. Thus, the composition of both the groups was similar and independent of group allocation (χ^2^ = 2.260, *p* = 0.133), as expected by the random allocation of participants. No age-based statistically significant differences were found in the level to which the participants justified violence (*p* > 0.050).

### 3.2. Pre-Intervention Levels of Violence Justification

According to the data obtained on the AADS scale (where higher scores reflect lower violence justification), boys and girls showed different justification levels for physical violence (Figure 2). The main differences were observed regarding male and peer aggressors. Overall, boys showed greater justification of physical violence than girls (Figure 2). The most justified form of physical violence by both sexes was physical aggression (slapping) committed by a girl in response to previous verbal aggression (insult) committed by a boy, with 32.3% of the participants saying that they completely agree with this form of conflict resolution. The second most justified form of physical violence (31.1%) was physical aggression (pushing) committed by a girl in response to previous aggression (pushing) committed by a boy (specific data can be found in the Appendix A).

The baseline analysis of JVCT items (higher scores reflect a higher level of justification of violence) showed that the most justified female psychological aggression by girls and boys was ‘sulking or refusing to talk about issues with him/her’ (girls 84% vs. boys 78.1%) (Appendix A, respectively). Although certain female aggressions were slightly more justified by girls than boys, an analysis of the three dimensions of JVCT showed that female violence (Figure 3A) was, in general, more justified by boys. While verbal aggression and jealous tactics were justified equally, female dominance tactics were, specially, justified more by boys, and the difference was statistically significant (*p* = 0.032, r = 0.08).

Regarding male psychological aggressions, the most justified one was, as in the previous case, ‘sulking or refusing to talk about issues with him/her’ (girls 84.3% vs. boys 76.4%). Again, precise data are available in Appendix A. Focusing on JVCT dimensions, as with female violence, the data show a tendency towards greater justification of male aggressions on girls, (Figure 3B), although, as a whole, male violence is justified to an equal extent by boys and girls (Figure 3B).

Boys showed a slightly higher tendency to justify and accept psychological dating violence. Overall, boys were slightly more likely to justify female violence than male violence, and girls were likely to justify female and male violence equally. In general, our participants were more likely to be accepting of female psychological violence than male psychological violence.

### 3.3. Effectiveness of Educational Intervention about the Justification of Physical Violence

Analysis of AADS post-test scores showed an improvement in the justification of physical violence against partners, among both boys and girls. In boys, all dimensions were significantly improved, although statistically significant differences were found only in the female aggression dimension (*p* < 0.001 in the intervention group, r = 0.33), which implies that the intervention significantly helped the participants recognise this type of aggression. A significant improvement was also observed in the control group (*p* = 0.033, r = 0.09, Appendix A). Among girls, scores increased in the intervention and control groups, showing clear improvement and statistically significant differences in these dimensions: justification of female aggression against her boyfriend in the intervention group (*p* = 0.020, r = 0.21) and control group (*p* = 0.015, r = 0.04) as well as justification of male aggression (*p* = 0.005, r = −0.325) and aggression between peers (*p* = 0.001, r = −0.25) in the intervention group.

On analysing the effectiveness of the educational intervention on the level of justification of physical violence (Figure 4), no statistically significant differences were observed in either boys (*p* = 0.652) or girls (*p* = 0.977). While scores improved similarly in both of the groups and sexes, girls in the intervention group showed a greater decrease in the level of justification of male aggression and aggression among peers than those in the control group (Appendix A).

The difference-in-differences estimates comparing AADS dimensions before and after educational intervention are shown in Table 1. Treatment was associated with an increase of the female aggression score in boys; concretely, the DiD estimate showed an 0.5972 (standard error [SE]: 0.864)-point increase in this dimension. No significant changes were observed in the other dimensions. In girls, the intervention was associated with a 1.17 (SE: 0.945)-point increase in female physical aggression, as well as with a 0.81 (SE: 0.456)-point increase in peer aggression scores.

### 3.4. Effectiveness of Educational Intervention about the Justification of Psychological Violence

Analysis of the JVCT scores after the intervention showed an improvement compared to the initial values: all of the scores related to the level to which participants justify the use of psychological violence decreased. This improvement was more prominent in the intervention group, where both boys and girls (Figure 5) justified psychological violence against their partners to a lesser extent than before, and the differences were statistically significant.

In boys in the intervention group, statistically significant differences were found in all dimensions, with the level of total justification of psychological violence decreasing by 6.4 points (*p* = 0.031, r = −0.15). In girls, all the scores decreased in both groups (intervention and control), showing a clear improvement and statistically significant differences in most of the dimensions studied, especially in the intervention group. A notable decrease was seen in the level to which girls justify psychological aggression, both female and male, against their partners. When analysing the effectiveness of the educational intervention in girls, although the total score improved slightly more in the intervention group (−4.4 ± 17.0) than in the control group (−4.0 ± 11.3), no statistically significant differences were observed (*p* = 0.594) (Appendix A).

The evaluation of the difference-in-differences estimates showed a general decrease of the justification of female psychological violence in both boys and girls (Table 1). In boys, the higher effect was observed regarding female verbal violence, with an effect size of −0.88 (SE: 0.539) points. On the other hand, a higher effect was observed regarding female jealous tactics. When the data regarding male violence justification were evaluated, our analysis showed that, in both boys and girls, the dimension that changed the most was the verbal aggression.

## 4. Discussion

The present study evaluated the effects of an educational intervention on reducing the justification of dating violence as a method of conflict resolution among 14–18-year-old adolescent girls and boys. Such justification has been identified as one of the main risk factors for both perpetration of violence and victimisation [15,30].

Previous educational interventions based on a broad approach to preventing violence and promoting healthy relationships have been shown to have little effect on reducing teen dating violence [26], probably because of the multifactorial source of dating violence. In this line, interventions focused on reducing the risk factors involved in teen dating violence have been more effective in preventing and addressing teen dating violence, especially when implemented at an early age [10]. Therefore, educational interventions used to prevent dating violence should be focused on specific issues of teen dating violence. Few previous studies have analysed the justification of dating violence [28]. The few that do are almost exclusively focused on physical violence. Thus, as far as we know, no interventions have aimed at reducing violence in the settings discussed in this study.

Given our objective, our educational intervention has been effective. An improvement has been observed in the participants of the intervention group, as their acceptance and justification levels of violence, both physical and psychological, have decreased, which, according to evidence [18,31], should lead to a long-term decrease in the risk of dating violence perpetration and victimisation. Our study demonstrates that programmes focused on reducing violence-prone attitudes help address and eliminate violence among adolescents. Regarding physical violence, the levels of justification shown by our participants prior to the educational intervention indicate that boys, particularly, tend to justify aggression against a partner when it is committed by a girl. These results are consistent with those from a study that examined the justification of violence bidirectionally [32] and with more recent studies, which show higher levels of justification for physical violence committed by girls [33]. They are congruent with other reports [34,35] on the differences observed between boys and girls in their ratings of the severity of aggression, which suggests that participants may perceive that the same physical aggression is more harmful and has more serious consequences when the aggressor is a boy as well as when the victim is a girl [36,37].

Per this line of analysis, another reason why the participants justify female physical aggression to a greater extent is that, in physical aggressions, a girl may be more vulnerable and in a worse position to defend herself against the greater physical strength that a male aggressor may show [38]. This may lead the participants to show a greater level of empathy when the victim is a girl [39]. Hence, we believe that it would have been useful to have also measured the levels of empathy shown by our participants before and after the intervention, to know if there is any relationship with the levels of justification. We believe this aspect should be taken into account in future studies.

An improvement has been observed in the levels of justification of physical violence in both groups after the educational intervention, especially in the decrease of justification of female aggression against their partners. Although the levels of justification have decreased in the boys of our study, it is the girls who show lower levels of justification and in whom a greater improvement is observed after the educational intervention. These data are consistent with those obtained by Savasuk–Luxton et al. [40], in which the educational intervention reduced the acceptance of dating violence in girls but not in boys. This may be due to the tendency in girls to show higher levels of empathy towards the victim [41], which could favour greater awareness following an educational intervention and lead to a decrease in the level to which they justify violence.

It should also be noted that the efficacy of the educational intervention in the female intervention group was greater than that in the control group concerning the dimensions of justification of male aggression and peer aggression. These data are consistent with the current evidence, which supports the effectiveness of prevention programmes in reducing the risk of victimisation and perpetration of teen dating violence [13,42] and reinforces the existing tendency for boys to justify violence to a greater extent than girls [30,43].

Regarding psychological violence, before the intervention, the JVCT scores suggest medium-to-low justification, which is similar in boys and girls in both groups. These levels of acceptance of psychological violence can be explained by the difficulty reported in the literature, especially among adolescents, in recognising verbal aggressions, jealousy, and domination tactics as a form of violence in a dating relationship. Many young people do not interpret these acts as violent, or if they do, they tend to minimise them by downplaying their seriousness [43]. Our results are partially consistent with the literature, as some authors report a greater recognition of psychological aggression as a form of intimate partner violence by girls [44]. Likewise, discrepancies have also been observed in the literature as to whether it is girls [45] or boys [46] who justify psychological violence to a greater extent.

In terms of isolated aggressions most justified by both sexes individually, it worth highlighting that both boys and girls justify female psychological violence slightly more than male psychological violence, even in the case of the same psychological aggressions. These data are congruent with those obtained by Muñoz-Rivas et al. [28] in the validation study of the AADS and the JVCT with Spanish adolescents: girls reported higher levels of justification of jealous tactics. On the other hand, although the differences between the sexes are slight, the high levels of justification shown by participants for these behaviours are striking. These findings are consistent with the unanimous conclusions in the literature about the tendency observed in young people, especially girls, to show distorted ideas about love, to interpret jealousy as a sign of love [47] and to consider its absence as a sign of indifference and disinterest on the part of the partner [48] which, in turn, increases the risk of committing and suffering other forms of violence [49].

At the dimensional level, in the present study, boys and girls initially show the same levels of justification of female and male violence for jealous tactics and verbal aggression, with the latter being the most justified kind of violence. These results converge with those obtained by Bandera and Muñoz [50], who also reported that verbal aggression showed the highest levels. These results can be explained as a lack of the communication skills needed for non-violent coping and conflict resolution within the couple [51] as well as a possible distorted perception of these behaviours as an inherent part of the relationship or a demonstration of love [52]. Regarding dominance tactics, which are the least accepted, boys justify boys and girls equally, while girls justify male domination more than female domination, although to a lesser extent than boys. This may be based on the possible perpetuation of and belief in certain sexist attitudes and the greater resistance to change noted in boys, which is already reported by other authors who also observed higher levels of sexist behaviours and justification of violence in boys [53].

Regarding the relationship between the level of justification of psychological violence and having had a dating relationship, Fernández–González et al. [54] observed that students who had had a dating relationship justified dating violence to a greater extent, whereas our data indicate that participants who have been in a dating relationship have a slightly lower mean score in the JVCT dimension of domination tactics, both from girls towards boys and from boys towards girls, than those participants who have never had a partner.

Justification of violence changes with gender and age, along with the reasons for justification [55]. Both boys and girls in our study are more accepting of female physical violence than male physical violence as a method of defence against previous male aggression. Boys generally justify physical violence to a greater extent than girls, and the differences are statistically significant. Responses show that 36.9% of boys and 27.2% of girls ‘completely agree’ with a girl slapping a boy in reaction to a previous insult from him; similarly, 34% of boys and 27.8% of girls also justify a girl’s pushing of a boy who has previously pushed her. These data are consistent with the findings of Price et al. [56], who concluded that both boys and girls were more accepting of female violence than male violence, with boys offering a higher level of justification. Others [57] have also found that boys were more likely than girls to justify physical dating violence as a method of defence. Likewise, previous authors [58] concluded that girls justify their aggressions more as an outburst of anger (girls, 22.4% vs. boys, 13.9%), while boys do it to defend themselves from an earlier aggression (girls, 6.6% vs. boys, 13.0%).

The high levels of justification shown by participants for the use of violence as a form of defence against a previous aggression may be motivated by the beliefs that it is the only way to defend oneself and that it is fair to attack a previous aggressor, which often make aggression the preferred defence option for many adolescents [59]. This also highlights a lack of resources and social skills to solve conflicts in a non-violent way [60]. Misunderstood self-defence thus becomes a frequent cause of physical, psychological and sexual violence in relationships: individuals learn to defend themselves inaccurately by confusing it with revenge, which is returning the damage to the aggressor [61].

The level of justification of violence could be a modulating factor. As the level of justification of violence increases, so does the risk of both perpetrating and experiencing violence. Justification generates greater tolerance or acceptance of the received aggression in the victim, and there is a tendency to set fewer limits for the aggressor. The justification of violence favours its maintenance and chronification in the relationship dynamics, enabling both the victim and the aggressor to normalise it [31]. Thus, higher levels of justification considerably increase the risk of victimisation and perpetration of intimate partner violence. On the other hand, female physical aggression, even if mild, increases the likelihood of reactive male aggression [62]. Higher levels of justification for violence against girls [63] are put forward as a plausible reason for the higher levels of psychological violence observed in girls, both experienced and committed. For future research, it would be useful to measure participants’ perpetration and victimisation rates before and after the educational intervention to see if there are relationships with higher levels of justification.

Given the successful results obtained, it is worth highlighting the salience of this educational intervention: it is innovative in that it is focused on reducing the justification of violence and has proven to be an effective tool for adolescent health education, carried out by nursing professionals. It is also important because, according to the evidence, reducing justification helps reduce the risk of dating violence as well as its serious consequences to individual and community health, whether physical, sexual, psychological or social. These include mental health problems, dissatisfaction with life, depression, suicide attempts, low self-esteem, irascibility, anxiety, disruptive family environment, mood swings, addictions, poor academic performance, physical injury and unwanted pregnancy [64,65].

At this point, several drawbacks of the present study should be commented. All the participants are from the Murcia Region, which limits the extrapolation and generalisability of the findings to the rest of the country. The possibility of social desirability bias may have limited the veracity of the participants’ answers, and some procedural bias or information bias may have influenced the results as a result of participants’ answering of the tests automatically and quickly [66]. Furthermore, between the pre-test and post-test, it cannot be guaranteed that the participants of the control group were not exposed to any external dating violence prevention material or any training that could have affected the effectiveness of the intervention [67].

Another limitation of the study is related to the lack of assessment of the empathy levels of the participants, which would allow us to analyse its possible relationships with the levels of justification of dating violence. Therefore, future research should consider measuring the levels of empathy that participants show towards victims of partner violence, in order to obtain more consistent results.

Despite its limitations, this study has contributed to reducing the levels of justification of dating violence among adolescents from the Region of Murcia. It provides updated knowledge that allows us to deepen our understanding of the attitudes and factors that favour the initiation and maintenance of violence, as well as to design and implement more effective and appropriate prevention programmes through school settings. Future interventions should promote the rejection of any presentations of dating violence and provide the tools and skills for non-violent conflict resolution.

## 5. Conclusions

This study confirms that our educational intervention was successful and effective in reducing justification for violence, which, according to current literature, will be effective in reducing teen dating violence. The intervention was more effective in boys, especially regarding the reduction of female and male physical aggression as well as dominance tactics. The intervention also reduced violence justification in girls, but in this instance, differences with the control group were less evident, which reinforces the need to continue developing similar prevention strategies, but with more emphasis toward girls.

Educational interventions conducted by nurses, like the present one, may improve individual and community health by reducing dating violence. These interventions should be conducted with adolescents, especially from an early age, in order to provide them with the skills and resources necessary to avoid violence justification in couples. In this study, the levels of acceptance of psychological violence decreased more than those of physical violence, which suggests that more appropriate interventions are needed, especially among boys, to reduce the justification of violence in the physical sphere.

Preventing and avoiding teen dating violence is a complex process that requires educational programmes focused on reducing its causes and risk factors. Secondary or high schools will be the most appropriate setting for the nurses to implement the educational intervention. Prevention programmes should also involve parents and educators in order to increase their awareness and to ensure that the same educational model is followed both at home and at school, which helps increase the effectiveness of intervention in young people.

## Figures and Tables

**Figure 1 healthcare-11-01156-f001:**
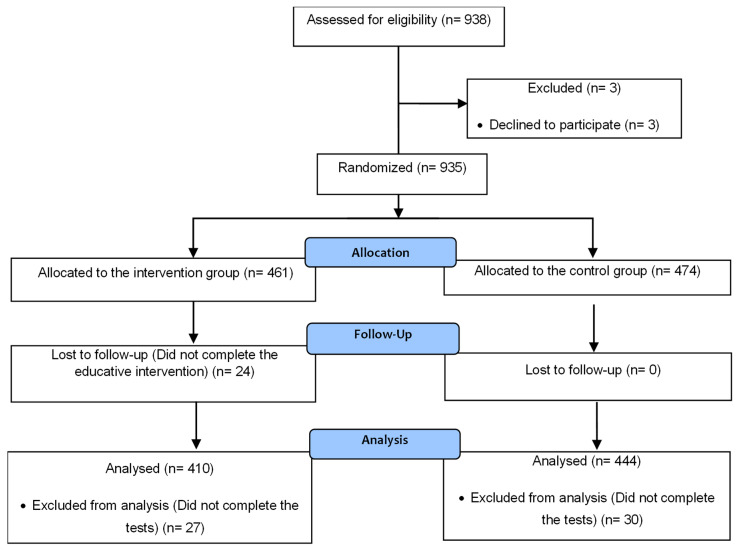
Distribution of the sample.

**Figure 2 healthcare-11-01156-f002:**
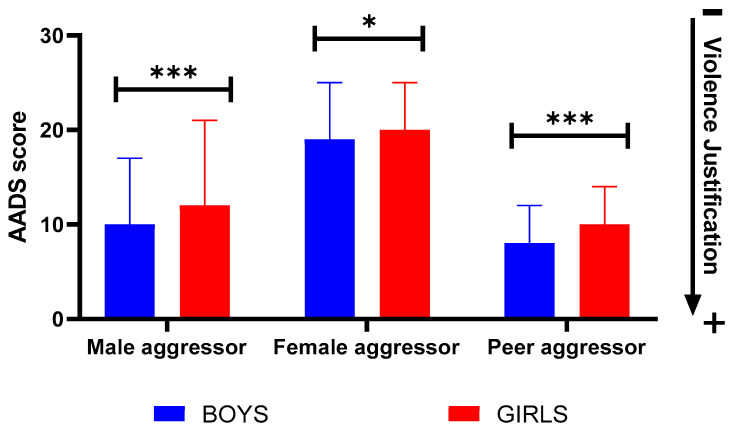
The level to which boys and girls justify, before the educational intervention, the physical violence committed, according to the sex of the aggressor, according to the AADS scale. Data represent the mean ± sd. Significant differences between groups were analyzed by Mann–Whitney *U*-test.* *p < 0.050*; *** *p < 0.001*.

**Figure 3 healthcare-11-01156-f003:**
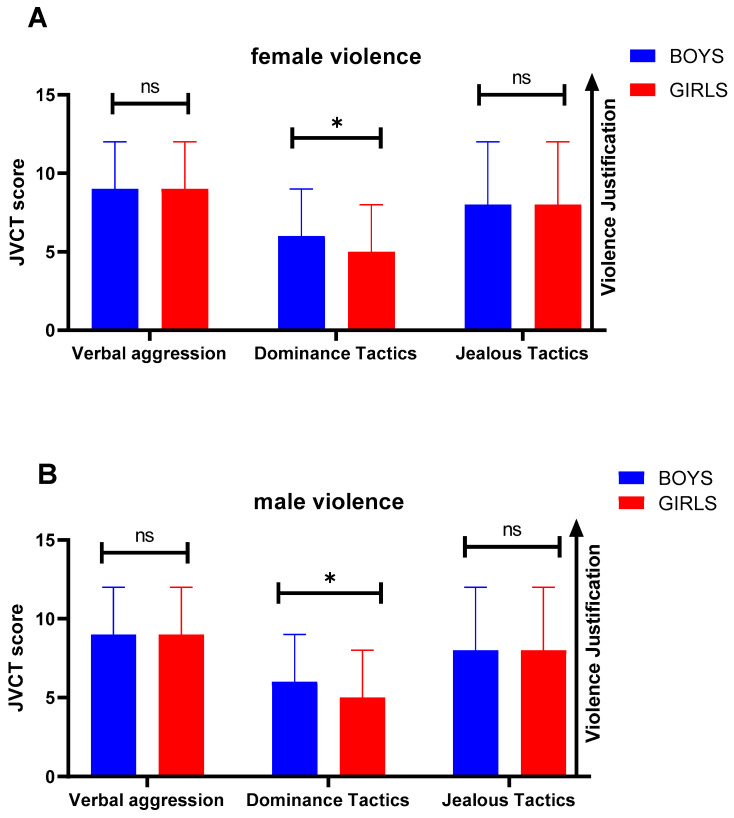
The level to which boys and girls justify female psychological violence (**A**) and male psychological violence (**B**), before the educational intervention, according to the sex of the aggressor, according to the JVCT scale. Data represent the mean ± sd. Significant differences between groups were analyzed by Mann–Whitney *U*-test. * *p < 0.050*; ns: no statistically significant differences.

**Figure 4 healthcare-11-01156-f004:**
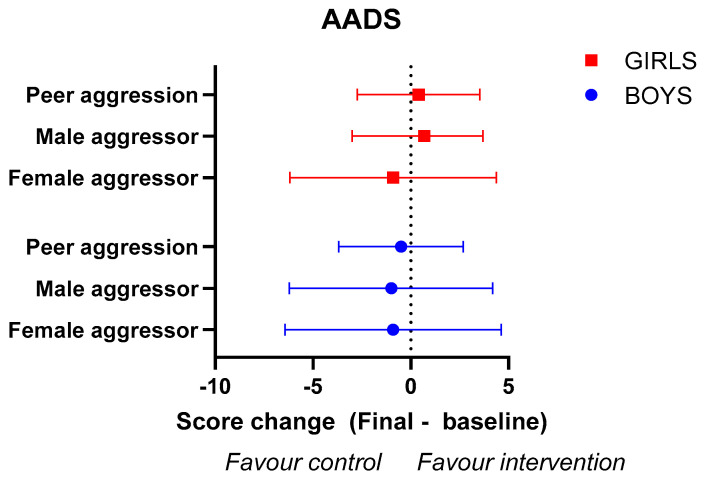
Analysis of comparative efficacy in boys and girls in the educational intervention group versus the control group, concerning the AADS scale. Data indicate intervention group changes (post-intervention − baseline) on the different AADS dimensions minus control group changes. Error bars indicate 95% CI of the change.

**Figure 5 healthcare-11-01156-f005:**
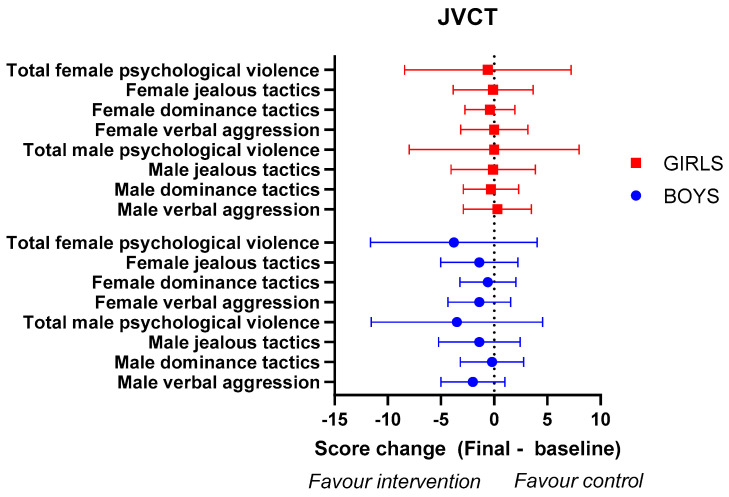
Comparative efficacy analysis between the boys in the educational intervention group versus the control group, concerning the JVCT scale. Data indicate intervention group changes (post-intervention − baseline) on the different AADS dimensions minus control group changes. Error bars indicate 95% CI of the change.

**Table 1 healthcare-11-01156-t001:** Post-intervention changes in AADS and JVCT dimensions for boys and girls in the intervention groups compared to the control group.

**BOYS**	**DiD Estimate**	**Standard Error**	**t Value**	**Pr (>|t|)**
AADS dimensions (physical violence)				
Female aggression	0.597	0.560	1.694	0.188
Male aggression	0.227	0.815	0.279	0.781
Peer aggression	−0.304	0.497	−0.612	0.541
JVCT dimensions (psychological violence)				
Female verbal aggression	−0.880	0.530	−1.633	0.103
Female dominance tactics	−0.253	0.508	−0.498	0.619
Female jealous tactics	−0.396	0.629	−0.630	0.529
Male verbal aggression	−0.465	0.524	−0.888	0.375
Male dominance tactics	−0.220	0.478	−0.459	0.646
Male jealous tactics	−0.445	0.634	−0.703	0.482

GIRLS	DiD Estimate	Standard Error	t Value	Pr (>|t|)
AADS dimensions (physical violence)				
Female aggression	1.169	0.645	1.237	0.117
Male aggression	0.043	0.729	0.059	0.953
Peer aggression	0.808	0.456	1.770	0.077
JVCT dimensions (psychological violence)				
Female verbal aggression	0.750	0.5407	1.387	0.166
Female dominance tactics	0.799	0.4471	1.786	0.075
Female jealous tactics	0.838	0.6196	1.352	0.177
Male verbal aggression	0.728	0.5476	1.330	0.184
Male dominance tactics	0.670	0.4938	1.357	0.175
Male jealous tactics	0.651	0.6322	1.030	0.304

The group × time difference-in-difference (DiD) indicator estimates the impact of educational intervention. Robust standard errors are also indicated. All regression models included variables for group (control or intervention) and time (baseline or post-intervention).

## Data Availability

The data presented in this study are available on request from the corresponding author.

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
