# Peer review of "Educational Intervention to Decrease Justification of Adolescent Dating Violence: A Comparative Quasi-Experimental Study"

_healthcare, 2023, doi:10.3390/healthcare11081156_

Round 1

Reviewer 1 Report

This paper evaluated the effectiveness of an educational intervention in reducing the justification of violence in adolescent dating. The work is of practice value, and it will be suitable for publication after addressing some issues.

1.      I enjoyed reading the introduction. The authors have structured the arguments well. Nevertheless, the authors stated that “it is imperative to conduct a nursing intervention in educational contexts”. I think it will be more convincing if the authors can cite some successful cases about nursing interventions.

2.      In the methods section, the Educational Intervention is the core of this study, so I think the author should give a more detailed description.

3.      In addition to presenting p-values, the authors had better add effect sizes in the statistical results.

I trust that these issues can be handled and recommend revising the paper.

Best wishes.

Author Response

Manuscript ID: healthcare-2289014: “Educational intervention to decrease justification of adolescent dating violence: a comparative quasi-experimental study”

REVIEWER #1

This paper evaluated the effectiveness of an educational intervention in reducing the justification of violence in adolescent dating. The work is of practice value, and it will be suitable for publication after addressing some issues.

R1#1. I enjoyed reading the introduction. The authors have structured the arguments well. Nevertheless, the authors stated that “it is imperative to conduct a nursing intervention in educational contexts”. I think it will be more convincing if the authors can cite some successful cases about nursing interventions.

We would like to thank the reviewer’s comments. We have included the following references in the revised paper to reinforce our statement.

  • Peterson, K., Sharps, P., Banyard, V., Powers, R. A., Kaukinen, C., Gross, D., Decker, M. R., Baatz, C., & Campbell, J. (2018). An Evaluation of Two Dating Violence Prevention Programs on a College Campus. Journal of Interpersonal Violence, 33(23), 3630–3655. https://doi.org/10.1177/0886260516636069
  • McFarlane, Judith M.; Groff, Janet Y.; O'Brien, Jennifer A.; Watson, Kathy. Secondary Prevention of Intimate Partner Violence: A Randomized Controlled Trial. Nursing Research 55(1):p 52-61, January 2006.

R1#2. In the methods section, the Educational Intervention is the core of this study, so I think the author should give a more detailed description.

We completely agree with the reviewer and have included more information in this section to try to clarify the intervention.

R1#3. In addition to presenting p-values, the authors had better add effect sizes in the statistical results.

            As suggested by the reviewer, effect sizes were expressed as r-value, according to the American Statistical Association statement on P-values. The following reference was also included:

            Lakshmi Narayana Yaddanapudi. The American Statistical Association statement on P-values explained. J Anaesthesiol Clin Pharmacol. 2016 Oct-Dec; 32(4): 421–423. doi: 10.4103/0970-9185.194772.

I trust that these issues can be handled and recommend revising the paper. Best wishes.

Reviewer 2 Report

The central topic of the research article: Educational Intervention to Decrease Justification of Adolescent Dating Violence is of great interest although it has been approached from a very particular context and the results are not discussed with previous studies in this regard.

The introduction has few significant references. In some places it seems more like a manifesto or an opinion piece than an element of support for the arguments presented below. Once again, the topic is quite relevant, but it lacks more support and a better connection between the different sections of the document. In the introduction authors can explain their motivations and why the issue is relevant.

The literature review must be orderly: ideas are not connected. I suggest to group ideas and create a logic discourse.

Data collection and analysis must provide more insight in how the data was analysed. The justification of the methodology is not robust (too generic).This election must be explained properly with research criteria.

The paper must finish with a proper conclusions and not with a summary.

In general, the feeling is that there was no clear objective for this paper, since it does not clearly present a methodology for studying, neither the results of a study with s strong scientific support. My suggestions would be for the authors to describe carefully all the procedures undertook and deeply assess the conclusions and the validity of the study itself.

Author Response

Manuscript ID: healthcare-2289014: “Educational intervention to decrease justification of adolescent dating violence: a comparative quasi-experimental study”

REVIEWER #2

R2#1. The central topic of the research article: Educational Intervention to Decrease Justification of Adolescent Dating Violence is of great interest although it has been approached from a very particular context and the results are not discussed with previous studies in this regard.

The comment of the reviewer is fair interesting. Although it is true that many interventions have been carried out to reduce adolescent dating violence, the innovation of the present intervention was that it has focused on reducing the justification level of dating violence, not the violent act itself. For this reason, it is somewhat difficult to compare our results with other previous studies. Nevertheless, we have tried to include previous studies where justification of violence has been evaluated but without interventions, and other studies of interventions where levels of justification have not been considered. In our opinion, these modifications have served to clarify these aspects.

R2#2. The introduction has few significant references. In some places it seems more like a manifesto or an opinion piece than an element of support for the arguments presented below. Once again, the topic is quite relevant, but it lacks more support and a better connection between the different sections of the document. In the introduction authors can explain their motivations and why the issue is relevant. The literature review must be orderly: ideas are not connected. I suggest to group ideas and create a logic discourse.

We agree with the reviewer comment. We have revised bibliographic references, and have adjusted the introduction to better state our motivations.

R2#3. Data collection and analysis must provide more insight in how the data was analysed. The justification of the methodology is not robust (too generic).This election must be explained properly with research criteria.

We regret the lack of clarity in this regard. We have included several sentences and have also modified some paragraphs of these sections to clarify these issues.

R2#4 The paper must finish with a proper conclusions and not with a summary.

 Effectively, the conclusion of the original paper was clumsy. We have modified this section accordingly to better state the conclusions of our research.

In general, the feeling is that there was no clear objective for this paper, since it does not clearly present a methodology for studying, neither the results of a study with s strong scientific support. My suggestions would be for the authors to describe carefully all the procedures undertook and deeply assess the conclusions and the validity of the study itself.

We thank the reviewer’s comments. We have tried to follow his/her indication in consequence.

Reviewer 3 Report

The reviewer examined the submitted paper entitled “Educational Intervention to Decrease Justification of Adolescent Dating Violence: A Comparative Quasi-Experimental Study” with high interest. In the submitted paper, the effects of educational intervention on adolescent dating violence were examined from data from different six schools in Spain. The author effectively used a quasi-experimental method to specify the causal relation between the targeted variables. Consequently, the author succeeded in observing the statistically significant and positive effects of the educational intervention on adolescent dating violence.

As the authors pointed out, this study had some limitations. The authors measured attitudes toward aggression in dating situations based on a questionnaire; therefore, the responses of the students might be affected by the socially desirable bias. By comparing changes in attitudes between the intervention and control groups, the reviewer believes that the authors could avoid this problem using a method of causal inference. The reviewer highly recommends this point.

However, the reviewer has some other concerns regarding the submitted paper, which are mainly methodological or technical points. The reviewer believes that the submitted paper should be revised based on these points. If the authors disagree, they may accordingly address the reviewer’s concerns. If the authors are able to convincingly address the concerns of the reviewer, their paper will be recommended for publication.

First, the authors should correctly explain what the error bars in the figures indicate. In Figures 2, 3, 4, and 5 even though the graphs have some error bars, the authors do not explain their significance. The reviewer wonders whether they are standard deviations, standard errors, 95% confidence intervals, or others.

The reviewer believes that readers’ impressions of the analytical results of the authors’ study partly depend on what the error bars indicate. Additionally, the reviewer believes if these error bars indicate standard deviation, the authors should use the error bars indicating standard errors or 95% CI to express the analytical results of the authors’ study. By doing so, readers can see the statistical significance of the analytical results more clearly.

Similarly, the reviewer believes that the graphs presented in the supplementary material can be improved. Boxplots were used to demonstrate the analytical results. However, these box plots are not helpful for readers to determine the statistical significance of the analytical results. The reviewer suggests that the authors should use error bars (se) or 95% CI to express the analytical results.

Next, the authors compared the analytical results based on the Wilcoxon test between the interventional and control groups. Nevertheless, most analytical results based on the Wilcoxon test are not demonstrated in the manuscript, and they are only presented in the supplementary material. Consequently, readers must look for relevant information in the supplementary material to check the statistical validity of the analytical results referred to in the submitted paper. The reviewer suggests that the readability of the submitted paper could be improved by fixing the construction. The Discussion section is unnecessarily long, in which the same contents were repeatedly referred to. The reviewer believes that the authors can shorten the Discussion section without substantial information loss and demonstrate the analytical results of their study more effectively.

Lastly, the authors simply compared the attitude scores between the pre- and post-experiment for each group (intervention or control group). However, the reviewer believes that the authors could use difference-in-differences analysis to estimate the effects of an educational intervention on adolescent dating violence. Moreover, by adopting difference-in-differences analysis, the authors could examine not only differences in statistical significance but also differences in effect sizes. Consequently, the submitted paper will be more persuasive to readers. Therefore, the reviewer hopes that the authors will examine the possibility of adopting a difference-in-differences analysis.

Thank you for the opportunity to review this paper. The reviewer sincerely hopes that his comments will contribute to the improvement of the authors’ study.

Author Response

Manuscript ID: healthcare-2289014: “Educational intervention to decrease justification of adolescent dating violence: a comparative quasi-experimental study”

REVIEWER #3

The reviewer examined the submitted paper entitled “Educational Intervention to Decrease Justification of Adolescent Dating Violence: A Comparative Quasi-Experimental Study” with high interest. In the submitted paper, the effects of educational intervention on adolescent dating violence were examined from data from different six schools in Spain. The author effectively used a quasi-experimental method to specify the causal relation between the targeted variables. Consequently, the author succeeded in observing the statistically significant and positive effects of the educational intervention on adolescent dating violence. As the authors pointed out, this study had some limitations. The authors measured attitudes toward aggression in dating situations based on a questionnaire; therefore, the responses of the students might be affected by the socially desirable bias. By comparing changes in attitudes between the intervention and control groups, the reviewer believes that the authors could avoid this problem using a method of causal inference. The reviewer highly recommends this point.

We sincerely appreciate the reviewer's comments, which undoubtedly denote her/his extensive knowledge of this type of studies. However, although the comments are relevant, we consider that they are outside the scope of this study. Considering the presence of a control group, we assume that the changes in the intervention group compared to this group, as shown in figures 4 and 5 of the original article, are more clarifying, as has been done in other previous studies carried out carried out by us and other authors [1, 2]. However, it is true that the inclusion of other analyses, such as the suggested differences-in-differences analyses, could be of some interest, which is why they have been included in the reviewed article.

References:

  1. Blackman A, Foster GD, Zammit G, Rosenberg R, Aronne L, Wadden T, et al. Effect of liraglutide 3.0 mg in individuals with obesity and moderate or severe obstructive sleep apnea: The scale sleep apnea randomized clinical trial. Int J Obes. 2016;40:1310–9.
  2. Galindo Muñoz JS, Morillas-Ruiz JM, Gómez Gallego M, Díaz Soler I, Barberá Ortega M del C, Martínez CM, et al. Cognitive Training Therapy Improves the Effect of Hypocaloric Treatment on Subjects with Overweight/Obesity: A Randomised Clinical Trial. Nutrients. 2019;11:925.

However, the reviewer has some other concerns regarding the submitted paper, which are mainly methodological or technical points. The reviewer believes that the submitted paper should be revised based on these points. If the authors disagree, they may accordingly address the reviewer’s concerns. If the authors are able to convincingly address the concerns of the reviewer, their paper will be recommended for publication.

R3#1.- First, the authors should correctly explain what the error bars in the figures indicate. In Figures 2, 3, 4, and 5 even though the graphs have some error bars, the authors do not explain their significance. The reviewer wonders whether they are standard deviations, standard errors, 95% confidence intervals, or others.

We regret the lack of information on this regard on the original paper. This information has been included on figures 2-5.

R3#2- The reviewer believes that readers’ impressions of the analytical results of the authors’ study partly depend on what the error bars indicate. Additionally, the reviewer believes if these error bars indicate standard deviation, the authors should use the error bars indicating standard errors or 95% CI to express the analytical results of the authors’ study. By doing so, readers can see the statistical significance of the analytical results more clearly.

Again, we consider that the information has been included in the previous point.

R3 #3- Similarly, the reviewer believes that the graphs presented in the supplementary material can be improved. Boxplots were used to demonstrate the analytical results. However, these box plots are not helpful for readers to determine the statistical significance of the analytical results. The reviewer suggests that the authors should use error bars (se) or 95% CI to express the analytical results.

 We fully agree with the reviewer. In fact, data represented in the boxplot referred to the 95% CI. We have included this information in the figures of the supplementary material to clarify this issue following the reviewer indications.

#4- Next, the authors compared the analytical results based on the Wilcoxon Test between the interventional and control groups. Nevertheless, most analytical results based on the Wilcoxon test are not demonstrated in the manuscript, and they are only presented in the supplementary material. Consequently, readers must look for relevant information in the supplementary material to check the statistical validity of the analytical results referred to in the submitted paper. The reviewer suggests that the readability of the submitted paper could be improved by fixing the construction.

At this point, we have some doubts about the reviewer’s comment. To the best of our knowledge, main results derived from the study are shown in the original version of the paper. In addition, p-values, as well as r-values to show effect sizes have been included. Supplementary material has been defined as relevant information that does not form part of the main body of the paper, which may include additional data, large tables, additional figures or appendices. Therefore, we consider that the inclusion of the supplementary material within the body of the article could make it too long and could distract attention from the main results obtained.

R3#5.- The Discussion section is unnecessarily long, in which the same contents were repeatedly referred to. The reviewer believes that the authors can shorten the Discussion section, without substantial information loss and demonstrate the analytical results of their study more effectively.

 We fully agree with the reviewer. We have modified the discussion section accordingly to try to adjust to the reviewer’s indications.

#6- Lastly, the authors simply compared the attitude scores between the pre- and post-experiment for each group (intervention or control group). However, the reviewer believes that the authors could use difference-in-differences analysis to estimate the effects of an educational intervention on adolescent dating violence. Moreover, by adopting difference-in-differences analysis, the authors could examine not only differences in statistical significance but also differences in effect sizes. Consequently, the submitted paper will be more persuasive to readers. Therefore, the reviewer hopes that the authors will examine the possibility of adopting a difference-in-differences analysis.

We agree the reviewer comment. We have done such analysis and included the associated information in the results section. We believe that performing such analysis has improved the quality of the paper, as suggested by the reviewer.

Round 2

Reviewer 3 Report

Thank you for your effort to address the issues pointed by the reviewer. The reviewer confirmed that the issues referred by the reviewer were adequately addressed by the authors. The reviewer would recommend the revised paper for publication.